# Cellular Models of Alpha-Synuclein Aggregation: What Have We Learned and Implications for Future Study

**DOI:** 10.3390/biomedicines10102649

**Published:** 2022-10-20

**Authors:** Katrina Albert, Sara Kälvälä, Vili Hakosalo, Valtteri Syvänen, Patryk Krupa, Jonna Niskanen, Sanni Peltonen, Tuuli-Maria Sonninen, Šárka Lehtonen

**Affiliations:** 1A. I. Virtanen Institute for Molecular Sciences, University of Eastern Finland, 70210 Kuopio, Finland; 2Neuroscience Center, University of Helsinki, 00014 Helsinki, Finland

**Keywords:** alpha-synuclein, SH-SY5Y, hiPSCs, organoid, aggregation, synucleinopathy, blood–brain barrier, microglia, astrocytes, overexpression, mutation, Lewy body, Parkinson’s disease

## Abstract

Alpha-synuclein’s role in diseases termed “synucleinopathies”, including Parkinson’s disease, has been well-documented. However, after over 25 years of research, we still do not fully understand the alpha-synuclein protein and its role in disease. In vitro cellular models are some of the most powerful tools that researchers have at their disposal to understand protein function. Advantages include good control over experimental conditions, the possibility for high throughput, and fewer ethical issues when compared to animal models or the attainment of human samples. On the flip side, their major disadvantages are their questionable relevance and lack of a “whole-brain” environment when it comes to modeling human diseases, such as is the case of neurodegenerative disorders. Although now, with the advent of pluripotent stem cells and the ability to create minibrains in a dish, this is changing. With this review, we aim to wade through the recent alpha-synuclein literature to discuss how different cell culture setups (immortalized cell lines, primary neurons, human induced pluripotent stem cells (hiPSCs), blood–brain barrier models, and brain organoids) can help us understand aggregation pathology in Parkinson’s and other synucleinopathies.

## 1. Introduction

For a scientist, entering the field of alpha-synuclein (α-syn) research is a daunting task. Since the discoveries in 1997 that mutations in the gene encoding α-syn (SNCA) result in an autosomal dominant form of Parkinson’s disease (PD) [1] and that it is prevalent in Lewy bodies [2], aggregates linked to disease, there have been over 10,000 original articles about this enigmatic protein (Source: Scopus, Figure 1). Additionally, α-syn also accumulates in Lewy bodies in disorders such as PD with dementia (PDD), dementia with Lewy bodies (DLB), and multiple system atrophy (MSA). These diseases are collectively termed “synucleinopathies” as they share this common pathological feature, whether the Lewy bodies are found in the cytoplasm, neurons, or glial cells. This review will focus mainly on PD, as the majority of studies attempt to model it. For a definite diagnosis of PD, Lewy bodies must be found in the substantia nigra pars compacta neurons in a post-mortem examination—although other neurons have Lewy bodies as well [3]. PDD presents similar motor symptoms to PD but with memory loss and Lewy bodies in cortical areas, and DLB is also similar to this pathologically [4]. Lastly, MSA has the unique feature that the Lewy bodies are found in the oligodendrocytes [5]. Why are these diseases, all containing neuropathological deposits of α-syn, still somewhat distinct from one another?

α-syn is expressed in several cell types in the central nervous system (CNS) as well as in the periphery in humans [6]. As a monomer, α-syn forms a heterogenous population of different states in aquatic solution, which is mostly due to the fluctuating C-terminal tail and heavily dependent on the ligands it can interact with and the environment [7,8,9,10]. In terms of amino acid sequence, the first 25 amino acids are important, with the first 12 amino acids being crucial for lipid binding and docking (Figure 2). In the second α-helix are the hydrophobic areas (non-amyloid component, NAC) that are important for the oligomerization and aggregation of α-syn [11,12,13]. α-syn is often referred to in the literature as an “intrinsically disordered” protein, but it is rarely explained why this is important. Intrinsically disordered proteins have no unique 3D structure, which allows for greater conformational flexibility, results in a larger surface area for interacting partners, and comes with more post-translational modifications [14]. These properties are considered advantageous for the protein’s function in the cell, giving it both tight control as well as the ability to quickly react to changing conditions. However, one can guess that this dynamic nature results in difficulties in wrangling intrinsically disordered proteins as drug targets. α-syn is considered mostly unstructured and belongs to a family of amyloidogenic proteins. Additionally, one more layer of complexity to the folding properties of α-syn is the recent demonstration of its association with membrane-less organelles. The association of intrinsically disordered proteins with membrane-less organelles could provide a possible explanation for the triggering of the amyloidogenesis pathway in multiple neurodegenerative diseases [15]. While in physiological conditions, the formation of α-syn-related liquid-liquid phase separation is not favored, mutations, pathological stressors such as metal ions (Cu^2+^, Fe^3+^), or high local concentrations of α-syn [16] may facilitate it. Additionally, this unstructured region of α-syn is converted to β-sheet-rich structures that form fibrils [17]. In short, misfolding of the soluble monomer of α-syn to temporary oligomeric structures and then into the ordered fibrillar structure is considered to be the process of aggregate formation. Further, formed aggregative structures and shapes may change based on the post-translational modifications and, importantly, depending on the experimental set-up (refer to graphical abstract).

It is also important to note here that for the purposes of this review when taken out of context (i.e., not associated with a particular study), we refer to monomers as the 14 kDa native protein, oligomers as larger species of α-syn that may or may not be toxic, and fibrils as the above-mentioned β-sheet structure. It is not always clear in the papers discussed below what an oligomer vs. a fibril is in some cases, particularly when taking into consideration recent research into antibodies used to label these conformations [18].

## 2. Modeling Alpha-Synuclein Aggregation in Non-hiPSC Lines: What We Can Learn and Caution with Outcome Measures

Since α-syn is not naturally present in non-vertebrates, studying the propensity of this protein to aggregate has relied heavily on cellular models. As such, numerous cellular models have been generated to broaden our understanding of the role this protein has on the cellular dysfunctions observed in the disease. These models are based on the overexpression of either wild type (WT) or mutated α-syn or the addition of an exogenous protein. Multiple cell lines have been utilized to study α-syn aggregation and toxicity, ranging from differentiated immortalized cells to primary neurons of mammalian species. Aggregation can be achieved by overexpression of mutated α-syn and multiplications in the SNCA gene as well as by using exogenous α-syn preformed fibrils (PFFs) or by patient-derived aggregates. Ideal models of α-syn capture the key pathological events as seen in the clinics, such as the spread of aggregated α-syn, preferably in the form of Lewy bodies, and dysfunction of modeled cells.

There is a high amount of evidence supporting the causative role of α-syn in synucleinopathies, which has been derived from the discovery of numerous autosomal dominant SNCA mutations and multiplications [1,19,20,21,22]. Since then, these mutations have been discovered to have effects on the structure and aggregation dynamics of α-syn [23,24,25]. To summarize, these mutations alter the disease phenotype, severity, and age of onset, but the exact mechanism behind these aspects is still elusive. Despite this, the known mutations, such as A53T, A30P, E46K, H50Q, and G51D have been successfully used to model synucleinopathies both in vitro and in vivo.

With hundreds of studies using different cell lines, forms of α-syn, and outcome measures, it makes wading through the literature a difficult undertaking for a person new to the field. With this section of the review, we aim to summarize and highlight studies of interest in the field using immortalized and primary cells. Table 1 shows several recent studies that add to this but is not inclusive of all studies.

Several cellular studies have demonstrated the capacity for overexpressed or mutated α-syn to cause neuronal dysfunction as seen in PD, as follows: disruption of the ubiquitin-dependent proteolytic system in PC12 cells (an immortalized cell line of rat pheochromocytoma often used in neuroscience research [61]) with the A53T mutation [62], disruption in calcium signaling in SH-SY5Y cells with the A53T mutation [29], impaired dopamine (DA) release in stable PC12 cells expressing non-toxic amounts of A30P mutant or WT α-syn [63] as well as vesicular DA storage impairments in a human mesencephalic cell line (MESC2.10) expressing A53T mutant α-syn [64]. In addition to these immortalized cell models, murine primary neurons have also been broadly used to study the effects of α-syn overexpression. In a rat midbrain DA primary neuron culture, A30P and A53T mutations have been shown to have a decreasing effect on the regenerative capacity of the neurons [65]. Furthermore, primary cortical neurons of mice expressing A53T mutant α-syn have displayed reduced mitochondrial mobility as well as a decrease in mitochondrial membrane potential and in respiratory capacity [66]. There are also multiple different approaches to achieving expression of mutated *SNCA* or overexpression of the WT version. A few examples include using plasmid constructs [67,68] or viral vectors [69].

While these studies and others contribute important findings to the field of PD research, particularly in terms of using mutations that confer the disease. These mutations often have a more severe phenotype than an expression of the WT, which matches the clinical presentation of, for example, the A53T mutation, which leads to an earlier onset and more severe progression of the disease [70]. However, overexpression/mutation of α-syn may not be the best way to model sporadic disease in cell monoculture. DA neurons in particular are known to be sensitive to stressors [71], and therefore an abnormal level of protein expression at an acute timepoint would likely cause dysfunction regardless of the protein being expressed. Sure enough, already more than 20 years ago, researchers showed that green fluorescent protein (GFP), commonly used as a reporter and/or control in studies modeling neurodegeneration, can be toxic to cell lines with GFP overexpression [72] and also to primary cortical neurons [73,74]. Therefore, caution needs to be taken in interpreting the results from cell studies where α-syn is overexpressed. Which control protein or mutation was used to compare the effects to? What was the level of gene expression of α-syn versus the control protein/mutation? These questions need to be answered before one can conclude that the cell death or dysfunction was the result of α-syn specifically. 

In contrast to overexpression/mutation models, the following one major advance has emerged in recent years in the field of α-syn research: in 2009, Luk et al., were able to demonstrate that the addition of α-syn PFFs to SH-SY5Y culture (an immortalized cell line of human neuroblastoma, often used as a neuronal model, including as a PD model since DA markers have been observed [75]) induced recruitment and phosphorylation of endogenous α-syn into insoluble pathogenic forms, which leads to the formation of Lewy body-like intracellular inclusions. In most studies using α-syn PFFs the protocol to generate the fibrils is similar where recombinant mouse or human α-syn monomer is purified, put at 37 °C, and shaken for 7 days. This results in the fibrillar form of α-syn which is then sonicated (either using a probe tip or water bath sonicator) and applied to the culture [76].

Similar findings have also been observed in primary neuron cultures seeded with exogenous α-syn fibrils [77,78]. In addition, these studies reported synaptic dysfunction and cell death induced by the Lewy body-like inclusions. Findings from α-syn PFF cellular models have raised questions about whether the exogenous pathological forms of α-syn could induce synucleinopathies in vivo as well. Remarkably, brain injection of α-syn PFFs is sufficient to cause cell-to-cell transmission of pathological α-syn, leading to Lewy body-like inclusion formation in A53T α-syn transgenic [79] and in non-transgenic mice [80,81], and countless others). More recently, seeded α-syn PFFs induced the aggregation of endogenous α-syn while significantly increasing the secretion of nanoscopic α-syn aggregates by disrupting the protein homeostasis of SH-SY5Y cells [31] and in Cascella et al. (2021) [82], where they used rat primary hippocampal cells and SH-SY5Y cells (as well as hiPSC-derived DA neurons) to inspect the spread of differently produced α-syn fibrils/oligomers and outer lipid membrane dysfunction and intake of oligomers. Their findings show that by comparing two different α-syn fibril oligomeric structures, one with a disordered secondary structure and the other with a β-sheet rich core, the latter can cause major membrane disruption and, in their model, be toxic to the neurons, whereas the former one is biologically inert.

Although the α-syn PFF model is not perfect, as one is adding exogenous material that is produced in *E. coli* or in HEK293 cells, it has strongly demonstrated the possibility of endogenous α-syn aggregation as seen in patients. This can be used to model different synucleinopathies by adding the PFFs to different cell types and observing the effects. Of course, researchers also need to take into consideration the amount of α-syn PFFs added—adding a too-high concentration and causing unfettered cell death, but too low concentration may not result in any significant event. Additionally, there may be batch-to-batch variation between produced α-syn fibrils, as well as differences in sonication steps [83]. This can affect the length of the α-syn fibrils and consequently their seeding efficiency [84]. Therefore, a standardized protocol would be best suited so that studies between labs could be compared side by side. There are good protocols for producing α-syn fibrils [78] or using them in vivo [83,85], which could serve as starting material for 2D cultures as well, if researchers were to agree on the concentration of PFFs and the outcome measures used.

Another important factor to be taken into consideration while assessing the relevance of an α-syn PFF model is that the Lewy body pathology is sometimes achieved with experimental manipulations such as additional factors assisting the intake of fibrils [76]. This is in contrast with the majority of PD cases since they are sporadic and express normal levels of α-syn. That being said, the ease of access and biological relevance of the α-syn PFF model in relation to both sporadic and genetic PD is high since it can recapitulate the uptake and seeding of α-syn in WT cells as well as those with overexpressed or mutated α-syn. Here, in particular, the propagation of α-syn is worth mentioning, since this has long been considered a key event in sporadic PD [86,87]. Interestingly, both α-syn PFFs, as well as tau fibrils, have been shown to be taken up in a similar manner in mouse-derived multipotent neural stem cells [88] and in rat neuroblastoma cells and oligodendrocyte-like cells [89].

One option for α-syn fibrils to become internalized and at the same time propagate from cell to cell that has been shown recently is the tunneling of nanotubes between cells. Work from a mouse brain tumor, neuron-like CAD cells, and mouse primary cortical neurons to hiPSC-derived models has shown these tracks, or at least the close proximity of acceptor and donor cells, between cells to be a potential spreading mechanism for α-syn fibrils [90,91,92]. These papers also demonstrate the importance of lysosomal-mediated vesicle propagation of α-syn fibrils in neuronal-like CAD cells. These results are fascinating and relate strongly to intracellular lysosomal degradation pathways that have been shown to be crucial in neuronal cells. 

After the initial intake of α-syn fibrils, they have been shown to be transported into the neurons themselves. Transport can happen in the retrograde (towards cell soma) or anterograde direction, but with α-syn fibrils, the retrograde direction is more prominent, which has been widely demonstrated in mouse primary cortical neurons [93,94,95]. This is an advantage in terms of disease modeling since for the disease progression intracellular α-syn aggregation and the formation of Lewy body inclusions in the area of the cell is a key hallmark of PD and other synucleinopathies [96]. Specifically, in mouse primary neurons using α-syn PFFs the authors of [97] were able to verify that endosomal trafficking and lysosomal intake are the main events after PFF intake. While not the first observations of such events with primary neurons the above study shows exhaustive analysis of differentially conjugated or modified α-syn PFFs to determine important parameters regarding intracellular handling and the pathological events that follow. In the future, using a battery of analysis methods that differentiate between exogenously added α-syn fibrils and endogenous aggregated α-syn can serve to help researchers understand the aggregation process.

While in this review we discuss a lot about endogenous α-syn aggregation and how it can cause the formation of Lewy bodies, we do not get too deep into the topic and debate about the nature of Lewy bodies themselves. While it is important to point out that the term “Lewy body” is sometimes used indiscriminately in the literature; therefore, it would be in the best interest to have standardized ways to detect Lewy bodies in models of synucleinopathy. However, due to their heterogeneous quality from patient to patient or even cell to cell within a single cell line, the task has been difficult. Not a single antibody alone or three together are undisputed ways to prove the presence of Lewy bodies, although they may be convincing in a certain context. Usually, microscopes are limited by three different fluorophores when using fluorescent imaging and antibodies are commonly against ubiquitination, phosphorylated α-syn at ser129, and mitochondrial membrane fragment or lipids. So far, advances in imaging technologies and the adaptions to researchers’ needs have addressed this problem by both recognizing the Lewy bodies with classical antibody markers and the correlation of inclusion morphology analyzed by high magnification imaging, achieved e.g., by electron microscopy. One such approach is the correlative light and electron microscopy imaging (CLEM), demonstrated in the PD research field by [98], which reported the composition of Lewy bodies from PD patients. Later also [99] with CLEM and primary mouse hippocampal neurons or [100] with cryo-electron tomography and labeled α-syn fibrils in primary cortical neurons, were able to use these approaches while modeling the synucleinopathies with α-syn PFFs. These papers clearly demonstrate the benefit of characterization of the achieved Lewy bodies from the model used with multiple approaches, but it also puts pressure on small labs which need yet another validation assay for their α-syn-related paper to show biological relevance. Finding accessible ways to observe and categorize Lewy bodies is therefore also a priority in the synucleinopathy field. 

## 3. Modeling Alpha-Synuclein Aggregation in hiPSC-Derived Neurons: Opportunities and Challenges

Human induced pluripotent stem cells (hiPSCs) provide a unique platform for cellular modeling as they can be directly isolated from patients with disease and they maintain the genetic information of the donor, including chromosome abnormalities and gene mutations. This can be utilized when studying synucleinopathies since there are multiple known pathogenic mutations, as discussed previously. The majority of hiPSC models are focused on the triplication and A53T point mutation of SNCA due to their clear phenotype, both in the cell and in the patient (Table 2). However, SNCA multiplications in patients are rare [101], so this needs to be taken into account when applying findings to other forms of PD. HiPSC-derived DA neurons with SNCA triplication display elevated α-syn mRNA levels, which leads to increased expression of the protein itself [102]. Furthermore, the lines manifesting the SNCA triplication exhibit a decreased capacity to differentiate into DA neurons while also having lower neuronal activity when compared to control lines. HiPSC-derived neurons with this mutation have also shown increased levels of phosphorylated α-syn [103] as well as an increased accumulation of α-syn into aggregates, resembling Lewy body pathology [104]. In other studies using hiPSC-derived neurons [105] showed that overexpression of WT or pathological genetic variances of α-syn (A53T, A30P) can induce mitochondrial-related dysfunction, where WT and A53T were more potent to cause a difference, and A30P was not due to poor mitochondrial lipid layer binding. Similarly, [106] showed that other pathogenic variants which can produce oligomeric α-syn (E46K and E57K) and duplication of WT SNCA also impair mitochondrial anterograde trafficking in an hiPSC-derived neuronal model. This is in line with previously reported α-syn-related mitochondrial dysfunction as reviewed in [107,108,109] and seems to be through MAPK signaling, which is triggered by α-syn [110]. 

So far there are only a few publications where synucleinopathies are modeled in hiPSC-derived neurons with exogenous α-syn fibrils alone. Gribaudo et al., (2019) [118] demonstrated that aggregation of endogenous α-syn can be achieved in a hiPSC-derived cortical neuron model by the addition of either exogenous α-syn PFFs or ribbons, made as in Bousset et al., (2013) [136]. This work showed that while the intake of exogenous α-syn fibrils happens relatively fast, even within one day, the seeding of endogenous α-syn can take more than 30 days. In their model, they used human cortical neurons maturated from hiPSC-derived dorsal telencephalic neuronal progenitors. They showed that α-syn ribbons and fibrils possess different seeding properties, where ribbons are faster to cause prion-like propagation. They simultaneously showed the propagation of labeled α-syn fibrils through receptor cell axons towards acceptor cells using distal chamber microchips. An interesting result from the study is that these cortical neurons transport took α-syn fibrils at a velocity of ~2.6 µm/s intracellularly and the ribbons at a similar pace. The authors interpret the result of different strains propagating at a similar rate as an indication that the transport is mediated in the same way. Perhaps this method of measuring the propagation velocity of α-syn can aid in the estimation of the disease progression [137], although it is currently unclear what this means in terms of relevance to the patient. Lessons for future experiments can also be taken from the fact that they did not use any α-syn genetic variant cell line, but rather a moderately high α-syn concentration of 0.5 µM and were able to show mild accumulation of endogenously aggregated α-syn and Lewy body-like structures as follows: halo-like staining of phosphorylated α-syn, co-localizing with ubiquitination, p62-sequestosome1, and HSP-70. This pathological model was also shown to cause mitochondrial fragmentation and Ca^2+^ oscillation frequency changes, related to neuronal dysfunction, with ribbons being more potent. Previously, similar Ca^2+^ changes have been reported in hiPSC-derived neuronal models [138,139]. Recently, [140] witnessed similar higher potency with α-syn ribbons over fibrils in a screening study using hiPSC-derived DA neurons from multiple sources. In their approach, they used biologically relevant, more mature DA neurons (>45 days since differentiation), which were shown to be electrophysiologically active. Importantly, Tanudjojo et al. [140] were able to show PFF-induced endogenous α-syn aggregation in hiPSC-derived DA neurons in healthy control lines using 1 µM concentration. The authors used α-syn fibrils generated in *E. coli* and also observed that these fibrils amplify in the presence of brain homogenates derived from both PD and MSA patients. 

In the future, hiPSC-derived PD models can be used to introduce base-level questions by first conducting omic studies, which can then be answered in a more detailed study setup. Using hiPSCs to study α-syn aggregation has the advantage that, in addition to using human and, in some cases, PD patient-derived neurons, we can have a more biologically relevant model than immortalized or primary cells while keeping the tight experimental control present using cell lines.

However, despite the numerous advantages that patient-derived hiPSCs provide (pluripotency, supply of material, fewer ethical issues associated with human sample use), there are still limitations hindering their potential. One major issue is the heterogeneity of hiPSCs derived from different donors or even in clones from the same donor. It is also difficult to intervene with the variability as, in some instances, it can be either a protocol effect or a patient effect. Secondly, it is difficult to generate hiPSC-derived DA neurons with an aging phenotype since the reprogramming process will reset the cell into a more youthful state [141,142]. This is an issue when the goal is to model a disease that is heavily associated with aging. 

## 4. Effects of Alpha-Synuclein on Microglia

Lately, PD research has begun to focus on how the body’s immune system may contribute to disease pathogenesis [143]. This means not only looking at neurons, and in terms of CNS cells, microglia have emerged as an important study target.

The mouse microglia BV2 cell line has been extensively used for different microglial studies with α-syn. The BV2 cell line is generated by harvesting primary microglia from 1-week-old female mice (*Mus musculus*) and then transformed to a cell line by infecting the cells with retrovirus J2 carrying a v-raf/v-myc oncogene [144]. As mentioned before, α-syn can be found in monomeric, oligomeric, or fibrillar forms. The definition of oligomeric and fibrillar forms varies from paper to paper. To recapitulate an inflammatory state of microglia, lipopolysaccharide (LPS) and/or interferon-gamma (IFNγ) are used. LPS is an endotoxin that induces inflammatory activation of microglia, mouse microglia in particular, whereas IFNγ is an internal signaling protein that particularly stimulates human microglia.

The aggregation state of α-syn affects the activation of BV2 cells [51]. LPS (0.1, 1, 10 µg/mL) induces the release of cytokines TNF-α and IL-1β in a dose dependent-manner after 6 h incubation when inspected with enzyme-linked immunosorbent assay (ELISA). When BV2 cells are treated with different aggregation states and different concentrations (0, 0.1, 1, 2.5 µM) of α-syn, the fibrillic α-syn has the strongest effect on TNF-α and IL-1β excretion. TNF-α excretion hits a plateau at 1 µM dose, whereas IL-1β increases significantly only at the highest dose (2.5 µM). Monomeric α-syn induces a lower response, but interestingly, oligomeric α-syn fails to induce any activity. The aggregation state also affects α-syn internalization—monomers and oligomers are not taken into the cells as easily, whereas the fibrillar form gets phagocytosed. Depending on the concentration of α-syn fibrils, 60–80% of the cells internalized the fibrils. Moreover, the cells treated with α-syn monomers and fibrils appear to proliferate more (MTS-assay). 

Further to this, [145] tested several in vitro preparations of α-syn in BV2 cells. They also found that α-syn fibrils (compared to monomers, oligomers, and ribbons) showed the most robust release of TNF-α. Interestingly, they observed that several familial mutations in the monomeric form of α-syn also induced TNF-α release compared to the WT (A53T, A53E, E46K, H50Q, G51D). Conversely, when BV2 are primed with monomeric α-syn (for 2, 6, or 12 h), the microglia take an anti-inflammatory phenotype through ERK, NF-κB, and PPARγ pathways [49]. This appears to attenuate the pro-inflammatory and neurotoxic effects caused by fibrillic α-syn. 

In relation to microglial uptake of α-syn as mentioned above, using primary microglia cells isolated from an 8-month-old mouse brain, [146] demonstrated that the cells had decreased uptake of α-syn oligomers from the media of α-syn-overexpressing HEK293 cells compared to microglia from a 1 to 3-day-old mouse brain. 

Only a couple of studies have been published using hiPSC-derived microglia in vitro in concert with exogenous α-syn. Similarly to BV2 cells, hiPSC-derived microglia treated with oligomeric α-syn (high molecular weight, purified from *E. coli*) had a significant amount of Il-1β secreted compared to monomeric treated microglia, as well as caspase-1 activation [147]. In the other study, the authors co-cultured microglia with astrocytes to understand if there was a synergistic effect between the two cell types on aggregates [122]. Using α-syn fibrils (along with amyloid-β fibrils) they found that both cell types take up the extracellular α-syn, but that it appears the microglia were able to degrade it more efficiently than astrocytes. When the cells were in culture together, the microglia took up more α-syn over time versus the astrocytes; however, the total accumulation of the protein was reduced compared to the monoculture setup. This relationship between microglia and astrocytes and how they process α-syn aggregates needs further study.

With more and more research pointing to microglia as a major player in neurodegenerative diseases, it is important to explore their role in synucleinopathy. Current studies show interesting results in terms of α-syn causing an inflammatory response in microglia and also how microglia may interact with it and take it up or clear it away. However, much more research is needed as there are still central questions that remain unanswered as it is still unclear whether microglia play a causative role in PD or whether they are only reacting to α-syn accumulation and neuronal loss. 

## 5. Modeling Alpha-Synuclein Pathology Using Astrocytes

Astrocytes are the most abundant cells in the CNS, where they have many different functions [148,149]. They have an important role in the development of the CNS but also work as neuroprotective cells, providing both structural and metabolic support. In addition, they regulate the blood flow in microvessels and the permeability of the blood–brain barrier (BBB). They also contribute to the immune response via their cytokine and chemokine production [150]. In general, they possess other functions that are necessary for the normal function of the whole CNS.

A lot of knowledge of synucleinopathy in astrocytes has been achieved in the last two decades with animal in vivo and in vitro studies, but also in human in vitro studies. The development in hiPSC technology has provided a new source of human astrocytes for research use, which has made astrocyte studies more practicable and ethically less problematic compared to studies made with primary- or embryonic stem cell (ESC)-derived astrocytes. This development is clearly visible in studies, as most of the human astrocyte studies related to synucleinopathy have been made during the last few years and with hiPSC-derived astrocytes. 

As early as 2006, the effect of α-syn on astrocytes was studied with human primary astrocytes by [151] to see how exposure to α-syn affects the astrocytes. The study showed that human α-syn fibrils (though it is not explicitly mentioned or measured in the study) purified from *E. coli* (WT, and mutated forms A30P, A53T, and E46K) induces an inflammatory response in astrocytes. A-syn induced the expression of inflammatory mediator ICAM-1 and secretion of IL-6, and it was also shown that ERK1/2, JNK, and p38 pathways that were associated with the actions of α-syn, were activated in the presence of α-syn, and their inhibition decreased the IL-6 levels. Much later in 2021, Russ et al. [123] showed with hiPSC-derived astrocytes that α-syn fibril exposure leads astrocytes towards an antigen-presenting phenotype with upregulation of major histocompatibility complex and human leukocyte antigen molecules whereas TNF-α initiates a strong pro-inflammatory cascade with activation of nuclear factor κB (NF-κB) pathway and release of pro-inflammatory cytokines.

In 2017, the transfer of α-syn between astrocytes was studied by Rostami et al. [152] using hESC-derived astrocytes. They found out that after exposure of astrocytes to oligomeric α-syn, the astrocytes failed to degrade the oligomeric α-syn during the studied period even though the monomeric form was degraded rapidly. The accumulation of α-syn leads to swelling of the endoplasmic reticulum and mitochondrial disturbances. They also observed that stressed astrocytes transferred intracellular α-syn via tunneling nanotubes to healthy astrocytes (similarly to studies in neurons discussed above) and healthy astrocytes transferred mitochondria to stressed astrocytes in the same way. However, it must be kept in mind that the transfer of α-syn happens also via other ways.

To see how astrocytes differ in PD patients and healthy individuals, the differences between healthy and PD astrocytes with the LRRK2 mutation were studied by Sonninen et al. 2020 [119]. We compared hiPSC-derived astrocytes from three control lines, one isogenic control, and four PD lines with the LRRK2 G2019S mutation. The study showed that astrocytes from PD patients expressed higher levels of α-syn and possessed a more reactive phenotype compared to healthy astrocytes. The PD astrocytes also had altered calcium signaling and metabolism.

As mentioned earlier, astrocytes have neuroprotective functions in the CNS. These functions have been studied by Domenico et al. 2019 [121] and Tsunemi et al. 2020 [120] using hiPSC-derived DA neurons and astrocytes. The studies show that healthy astrocytes are able to protect the neurons (also neurons from PD patients) from neurodegeneration. However, in astrocytes from PD patients, the neuroprotective capability is compromised and may even induce neurodegeneration. In the study by Domenico et al. 2019 [121], they co-cultured the astrocytes and neurons from controls (from healthy individuals) and PD patients (from patients with the LRRK2 G2019S mutation). They found that in healthy neurons cultured with PD astrocytes, the neurodegeneration was clearly detectable when compared to cultures with control astrocytes. Importantly, the accumulation of α-syn was increased in neurons cultured with PD astrocytes. Instead, when PD neurons were cultured with control astrocytes, the astrocytes decreased the neurodegeneration of PD neurons compared to co-cultures with PD astrocytes. The results also show that in PD astrocytes, autophagy was impaired, leading to α-syn accumulation in the astrocytes. In the study from Tsunemi et al. 2020 [120], neuronal α-syn was observed to be cleared by astrocytes. However, the ATP13A2 mutations reduced the astrocytic uptake and clearance of α-syn leading to α-syn accumulation in DA neurons. They also noticed that ATP13A2 deficiency reduced phagocytosis and the endosomal pathway in both neurons and astrocytes.

Besides co-cultures with neurons, the co-culture with microglia and its effect on α-syn pathology has been conducted. As described earlier with microglia, the study from Rostami et al. (2021) [122] showed that co-culture of astrocytes and microglia increases the degradation of α-syn.

Human astrocyte models have provided a lot of information on the effects of α-syn on astrocytes. However, the research is still limited, and more knowledge must be gained with human cells. All of the studies reviewed above have used the 2D culture of astrocytes to study the effects of α-syn. The 2D culture models are easy to use, but they are not as physiologically relevant as 3D models, in which the cells are able to form an in vivo-like morphology. In co-culture models of astrocytes and neurons, both contact [120] and non-contact [121] cultures were used. In the in vivo environment, the astrocytes and neurons are in close contact with each other, and part of the signaling pathways requires close contact between the cell types. Thus, the contact co-culture setup mimics the situation in vivo better, but it might restrict the analysis that can be made. When modeling processes such as synucleinopathy, this becomes particularly relevant as the interplay between neurons and glia is important.

## 6. Role of the Blood–Brain Barrier in Alpha-Synuclein Pathology

Endothelial cells (ECs) form the walls of all the vessels in the body [153]. However, in the CNS, the microvasculature has specific properties that are not found in the periphery. In the CNS, the microvasculature restricts the movement of molecules due to a higher number of tight junctions and transporters. This property of microvasculature in the CNS is called BBB. In addition to the ECs, astrocytes and pericytes are needed to maintain and regulate the function of the BBB. The BBB is important for maintaining homeostasis in the CNS and protecting it from harmful substances such as inflammatory agents and toxins. However, the BBBs ability to not allow most molecules through also makes drug delivery to the CNS a difficult endeavor. 

Dysfunction of the BBB is associated with different neurodegenerative diseases such as Alzheimer’s disease, PD, Huntington’s disease, and amyotrophic lateral sclerosis [154]. However, its role in the progression of the disease is still largely unknown. Both in vivo and in vitro models have been developed to study the role of the BBB in α-syn pathology. In this part, we go through the research on the role of the BBB in α-syn pathology. As there is so little research on the BBB and α-syn using human cells, in addition to human in vitro models, we will also review the research made with in vivo and in vitro animal models. 

### 6.1. In Vivo Models

Several in vivo studies about α-syn and the BBB have been conducted to study different aspects of α-syn pathology. Some of them have concentrated on the transport of α-syn and others on the effect of α-syn on the BBB. Some of the studies have used in vitro models besides in vivo models to ensure findings and to further investigate the possible mechanisms behind the effect. 

The transport of α-syn has been studied from different perspectives by Sui et al., and Matsumoto et al., using in vivo models. In 2014, Sui et al. [155] studied the transport of α-syn through the BBB in mice. They showed that α-syn can be transported into and out of the brain by the BBB, and that the transport could at least in part happen via low-density lipoprotein receptor-related protein-1 (LRP-1). Later, Matsumoto et al. (2017) [156] also studied α-syn transport to the brain, but via extracellular vesicles (EVs). The study was conducted by injecting labeled α-syn-rich EVs from humans (PD patients and controls) into the periphery of mice. The study showed that the EVs are transported to the brain, especially with pre-treatment of peripheral LPS, and that EVs from PD patients trigger a stronger inflammatory response in microglia compared to EVs from controls. These results suggest that systemic inflammation might promote the access of α-syn containing EVs into the brain by initiating an inflammatory response that might also further enhance neurodegeneration. 

The effect of α-syn on the BBB in vivo has been studied with mice overexpressing human α-syn by Elabi et al., 2021 and Lan et al. 2021 [157,158]. In the study by Elabi et al., (2021), they observed that the BBB was compromised and pericytes were activated already at the early stages of the disease in mice with overexpression. Moreover, the density of microvessels was altered in mice with overexpression of α-syn. Instead, Lan et al., 2021 studied the mechanism leading to BBB disruption using an in vitro model in addition to an in vivo model. They saw a decrease in tight junction protein expression in ECs and an increase in BBB leakage caused by α-syn accumulation in and activation of astrocytes in vivo. Further in vitro studies showed that oligomeric α-syn induced BBB disruption mediated by the release of VEGFA and nitric oxide. The inhibition of the VEGFA signaling pathway protected the BBB from disruption in vivo and in vitro, so VEGFA seemed to be the primary mediator of BBB disruption.

Dohgu et al. (2019) [159] used an in vitro model using cells derived from animals to study α-syn pathology. They studied rat ECs and pericytes to elucidate how α-syn affects the BBB. They found that monomeric α-syn initiates the release of inflammatory mediators from pericytes, inducing BBB disruption. 

### 6.2. Human in Vitro Models

Only two studies [125,160] of the role of BBB on α-syn pathology have been made using human cells and even in these the other also used mouse primary cells. These two studies were conducted to study the effects of α-syn on the permeability of the BBB and the functions of different cell types.

In 2016, Kuan et al., studied the effect of α-syn PFFs on immortalized EC line hCMEC/D3 in monoculture and co-culture with mouse primary astrocytes or primary human fetal cortical cells. The study showed the impairment in the expression of several tight junction-related proteins in the presence of α-syn, a result also found in PD patient brains, but still, the barrier function of the ECs was not affected except transiently in neuronal co-cultures [160]. 

Research from Pediaditakis et al., 2021 tested the effect of a α-syn PFFs on an organ-on-chip model of the neurovascular unit. In the research, they used hiPSC-derived brain microvascular Ecs and DA neurons, primary human astrocytes, pericytes, and microglia. The study showed that exposure of the neurovascular unit model to α-syn leads to the death of DA neurons, activation of microglia and astrocytes, as well as disruption of the BBB, mainly observed by increased permeability. However, the study did not distinguish whether the disruption of the BBB was caused directly by the exposure of ECs to α-syn, or indirectly through other cell types [125]. 

In BBB research, the in vivo models are still important for studying CNS functions as the cell models are still relatively simple compared to tissues and whole organisms and thus do not recapitulate the complex functions found in vivo. However, the development of in vitro models has enabled more physiologically relevant modeling that mimics the in vivo situation better than earlier models. One example of this is the organ-on-chip model that was used in the above-mentioned study from [125]. Before the development of hiPSC technology, in vitro studies were mostly made with animal cells, as they were easily available, unlike human samples, the availability of which is very limited. HiPSC technology has provided an unlimited source of human cells, which has made in vitro studies with human cells more practicable. 

It is clear from these studies across different species and models that there is still a considerable amount to learn about synucleinopathy and α-syn aggregation related to the BBB. Models that use multiple cell types to understand how the BBB may be disrupted in disease are however bringing us closer to understanding this aspect of neurodegeneration.

## 7. Oligodendrocytes and Synucleinopathy

Oligodendrocytes are myelinating cells in the CNS. Defects in oligodendrocyte function can impair signal transduction due to the loss of the insulating myelin sheaths on the axons of the neurons and aggravate degeneration through the loss of neurotrophic factors. Collectively, these conditions are called demyelinating diseases, of which multiple sclerosis is the most common. The role of oligodendrocytes in synucleinopathies is not well understood, but emerging evidence has implicated their dysfunction as a possible factor in these diseases. As mentioned, MSA α-syn accumulates in oligodendrocytes and forms glial cytoplasmic inclusions (GCIs) [5], which has been recognized as a core pathological feature of the disease. Mature oligodendrocytes have not been found to express *SNCA*, leading to speculation that the GCIs are formed by exogenous α-syn being taken up by the cells. However, *SNCA* expression is transiently upregulated during oligodendrocyte maturation [161,162], leading to an alternative hypothesis that the gene may be dysregulated in MSA.

Oligodendrocyte progenitor cells (OPCs) are retained throughout adulthood. In the case of injury to the myelin sheath, OPCs can mature into myelinating oligodendrocytes and repair the damage. Importantly, it has been noted that the number of OPCs is increased in the brains of MSA patients. Despite the abundant OPC pool, there is considerable degeneration of myelin that is spatially associated with a high GCI burden [163]. Mice overexpressing human wild-type *SNCA* under the *MBP* promoter showed an increased number of OPCs in conjunction with myelin loss, mirroring the pathology found in brain tissue samples of MSA patients [164].

In vitro, overexpression of *SNCA* disrupted oligodendroglial maturation in the CG4 cell line [164], with similar findings reported in primary rat OPCs [165]. Transcriptomic profiling of OPCs derived from MSA and PD patient iPSCs has also shown impaired maturation into myelinating oligodendrocytes, with the phenotype shifting more towards an antigen-presenting cell type [126]. Contrary to previous studies using rat cells, Azevedo et al., found that *SNCA* overexpression increased the expression of maturation and myelination-associated genes in healthy iPSC-derived OPCs. However, the authors propose that α-syn expression may be necessary for early lineage commitment of precursor cells but impedes maturation at later stages [126]. 

While mature oligodendrocytes have not been found to uptake exogenous α-syn fibrils, OPCs and intermediate oligodendrocytes are competent. Primary rat OPCs treated with human α-syn PFFs develop intracellular inclusions containing endogenous rat α-syn, which are retained by the cell through maturation. Furthermore, α-syn PFFs induce the expression of endogenous α-syn in OPCs but not in mature cells [126]. Fibrillary α-syn perturbs normal oligodendrocyte maturation and has been shown to have cytotoxic effects if introduced at intermediate stages of development [126]. In healthy iPSC-derived OPCs, exposure to exogenous α-syn species resulted in the upregulation of genes associated with immune functions. In agreement with the findings by Kaji et al., (2020), the exposure also resulted in considerable toxicity that was more pronounced for fibrillar α-syn [126]. 

Whether oligodendrocyte dysfunction in MSA is the result of dysregulation of α-syn expression in maturating cells, uptake of aggregated α-syn by OPCs or a mix of both factors remains to be settled. While bona fide neuronal inclusions are rarely present in MSA, a study utilizing α-syn proximity-ligation revealed a high neuronal oligomeric α-syn load in MSA patient brain tissue [166]. The authors suggest that the oligomeric α-syn may have direct neurotoxic effects, while the uptake of neuron-derived oligomers by OPCs results in the characteristic GCI formation and demyelination. Still, further work is needed to understand the complex etiology of α-syn pathology in oligodendrocytes and how it may contribute to neurodegeneration in MSA and other synucleinopathies.

## 8. HiPSC-Derived Brain Organoids as an Emerging Modeling Platform for Synucleinopathies

Brain organoids have emerged as a promising new in vitro cell model for various neurological disorders. They are especially attractive for modeling developmental and familial diseases, as they have been shown to recapitulate human neurodevelopment and can be generated from patient-derived hiPSCs [167,168,169]. Directed differentiation of brain organoids enables the generation of more specific models for studying diseases affecting only certain brain regions or cell populations. In 2016, Jo et al., successfully generated midbrain-like organoids from hiPSCs, which could be used as PD models [170]. In addition to providing a human brain-like model system, brain organoids can be generated in large numbers for high-throughput experiments used in toxicity screening and drug development [171,172]. Additionally, they give the advantage of the surrounding cell types, not just DA neurons alone.

Along these lines, while brain organoids have been shown to contain a variety of ectoderm-derived neuronal and glial cell types, microglia are typically absent. During brain development in vivo, mesoderm-derived erythromyeloid progenitors migrate from the yolk sac and colonize the fetal CNS, giving rise to the adult microglia population [173,174,175]. Most protocols for deriving brain organoids include the use of dual-SMAD inhibitors to guide differentiation towards the neuroectoderm [176]. Although it is possible for microglia to emerge from residual mesodermal progenitors in unguided brain organoid differentiation, batch-to-batch variation in the distribution and numbers of microglia is a major limitation of this approach [177]. A possible solution to this is to derive the microglia progenitors separately and incorporate them into the organoids [178,179,180,181].

Brain organoids lack ECs and therefore do not develop vasculature. This limits the transfer of nutrients and typically results in a necrotic core, which can compromise neuronal survival and functionality. In addition, necrotic cells inadvertently activate microglia and astrocytes. Vascularizing brain organoids improves nutrient diffusion and allows for the construction of neurovascular models as well as studying BBB function. Transplanted brain organoids can be vascularized in vivo by invading vessels from the host [182], although the nonhuman origin of these vessels is a limitation. According to Pham et al., (2018), the vasculature can be humanized by incorporating human ECs into the organoids prior to transplantation [183]. Fully in vitro approaches include direct conversion to ECs [184], co-culture with human umbilical vein ECs [185], and assembloids composed of the brain and vascular organoids [186,187,188].

Toxin-based animal models of PD have been a standard in the field for decades. Still, they do not represent the true progression of the disease as follows: the loss of DA neurons is rapid and there is no synucleinopathy involved. It should be noted, however, that midbrain organoids can recapitulate the loss of TH+ neurons in vitro when exposed to toxic agents like MPTP or 6-OHDA [189]. While the relevance of these models to PD as seen in humans is debatable, the susceptibility of organoids to MPTP toxicity is an important finding. MPTP itself is not toxic and must be converted to MPP+ by the MAO-B enzyme, which is highly expressed in astrocytes. Recapitulation of this toxic cascade in midbrain organoids supports the notion that 3D cell models can mimic the physiological microenvironment of brain tissue.

As discussed earlier, mutations affecting the SNCA gene confer a high risk of developing synucleinopathy. Recently, Mohamed et al., (2021) generated midbrain organoids from hiPSCs carrying the SNCA triplication. They reported that these aged SNCA triplication organoids develop pS129 α-syn+ aggregates, which co-localize with both astrocytes and neurons. They also found the organoids to have decreased numbers of DA neurons [127]. Recent findings also suggest that combined SNCA overexpression and glucocerebridase deficiency can further promote the induction of synucleinopathy-like pathology in midbrain organoids [134]. While midbrain organoids are a natural choice for modeling PD, cerebral organoids could provide a platform for studying DLB and PDD, which share many of the genetic risk factors with PD. ApoE variants are known to affect Alzheimer’s disease risk, with the APOE4 isoform conferring a significantly increased risk. Interestingly, APOE4 has also been associated with the risk of DLB and PDD [190]. In an article published in 2021, Zhao et al., reported increased α-syn aggregation in ApoE^−/−^ cerebral organoids. Abnormal lipid metabolism and decreased glucocerebrosidase activity were also noted. Furthermore, it was shown that treatment with ApoE2 or ApoE3, but not ApoE4, could partially rescue the phenotype. Organoids generated from hiPSC lines homozygous for *APOE4* also showed increased levels of insoluble α-syn when compared to those derived from *APOE3* homozygote lines [132]. 

Mutations affecting the LRRK2 gene are the most common risk factor for developing familial PD. Furthermore, the presentation is often very similar to that of idiopathic PD. In 2019, Kim et al., showed midbrain organoids with inserted LRRK2-G2019S mutation to have lower gene expression of DA neuron markers TH, AADC, and DAT in comparison to the isogenic control. In addition, mutant organoids were reported to be more susceptible to MPTP toxicity. They also identified thioredoxin-interacting protein (TXNIP) as an intermediate actor in the pathology [131]. TXNIP has previously been reported to modulate α-syn accumulation via autophagy inhibition [191]. 

The current focus of hiPSC-derived midbrain organoid models for synucleinopathy has been rather narrow, with the effort being put into studying familial forms of the disease. With multiple genetic factors affecting disease risk, midbrain organoids carrying well-known risk variants have garnered the most interest. Indeed, there have been multiple reports of PD-like pathology arising organically in mutant organoids. In addition, some variants seem to increase the susceptibility of DA neurons to toxicity by MPTP or 6-OHDA, suggesting that these mutations can make DA neurons more vulnerable to stressors. In 2022, Rodriguez et al., showed that fibrillar α-syn is internalized by hiPSC-derived brain organoids and induces the formation of intracellular inclusions [133]. It should be emphasized, however, that no studies exposing midbrain organoids specifically to α-syn PFFs have been published yet. Furthermore, midbrain organoids intrinsically lack microglia. While there have been reports of microglia successfully incorporated into brain organoids [178,179], including a recent report on midbrain organoids [181], studies on α-syn processing and pathology in microglia-enhanced organoids have not been published.

Brain organoid technology has progressed rapidly in recent years. Improved culture systems, optimized differentiation protocols, and the development of new scaffold materials can help mitigate notable issues such as variability in represented cell populations and necrosis due to insufficient nutrient supply. More work is needed to create a truly comprehensive in vitro model of the human brain, including vasculature and microglia. There is a need for more focus on the function of microglia and how they interact with astrocytes and neurons and whether they contribute to the spreading of pathological α-syn in the brain. In addition, hiPSC-derived models of late-onset disease are inherently limited by the immature cell phenotype, which is not representative of the aged brain. This limitation was also highlighted in a recent review by Nogueira et al., (2022). They suggest that post-mortem human brain tissue samples or ex vivo organotypic culture should be used to validate the findings obtained from organoid models [192]. Nevertheless, with future advancements and standardization of methodology, brain organoids could provide a powerful platform for studying molecular and cellular disease mechanisms driving synucleinopathy in a biologically relevant, human-based system. 

## 9. Conclusions and Future

Although α-syn is involved in several diseases after thousands of publications, it is still not fully known how it contributes to disease initiation or progression. While α-syn is present normally in the human brain, not everyone develops neurodegenerative disorders such as PD. Additionally, while aging is a major factor in sporadic PD, it is likely that other factors such as inflammation, the environment, and genetics contribute as well. Therefore, studying α-syn in a single cell type in a dish (whether of human origin or not) without further perturbation will not unequivocally answer these questions. 

On the other hand, while studying the human brain in vivo using imaging techniques has given us useful knowledge, it is not always possible to gain detailed information on the pathophysiology of the disease. Additionally, obtaining patient post-mortem samples is not available to all labs, sample sizes are not large for rare diseases, and this only gives endpoint information. Here, cellular models have been instrumental in helping researchers understand the mechanism of α-syn aggregation and how it works in the cascade of disease initiation and progression. Immortalized cell lines and primary cells from rodents have been used robustly in the field, and with the advent of hiPSC technology, researchers hope to gain better insight into these human diseases. In particular, models that incorporate multiple cell types through co-culture, chips, or organoids are exciting for understanding synucleinopathies and other neurodegenerative diseases that can affect several cells and pathways in the CNS. With this comes some loss of experimental control, but their relevance to the human condition is higher than cell lines or animals alone.

In terms of α-syn itself, as discussed above, better models of Lewy bodies that would be observed in patients are necessary to bring the field forward. We have seen that there are several ways to model α-syn aggregation in vitro using a combination of techniques. Each of these has pros and cons, but we must always be clear and certain of what the outcome measures are and how the model is relevant to finding new therapies and understanding disease mechanisms. Perhaps currently, there is no “one size fits all” model of synucleinopathy. Furthermore, there can be a variety of types of aggregates, and whether and which aggregates are causing disease is still not fully clear. A recent study demonstrated that it is actually small soluble α-syn aggregates that cause toxicity to cells, and these resemble those found in the post-mortem patient’s brain [193]. A major question is also how a single protein is causing a spectrum of diseases. α-syn is a protein with different conformations and post-translational modifications, both also depending on conditions. Lately, there have been interesting publications studying conformations of α-syn using cryo-EM, where the authors, for example, have shown that α-syn filaments from MSA patients differ from those with DLB [194]. (See [195] for a review of recent research about different α-syn strains in relation to disease). This also has the potential to bring us toward more disease-relevant models. Importantly, not every PD patient develops Lewy bodies [196], nor does amount of α-syn-rich Lewy bodies always correlate with cell loss [197]. All of this leads to the idea that personalized medicine is the future of treatment for patients with synucleinopathy. Which kinds of aggregates are present and at what level is important if we are treating patients with pathology that likely began several years before symptoms appear. Here, hiPSCs are a powerful tool—cells can be grown from the patient and drugs can be tested on how they affect aggregation and cell function, for example.

Therefore, to conclude, using in vitro cellular models, regardless of source, is still a useful and viable way to test conditions that would be more time-consuming, expensive, and labor-intensive in vivo. The types of models discussed in this review all have their ways of contributing to further understanding of α-syn aggregation as long as we also take into account their limitations.

## Figures and Tables

**Figure 1 biomedicines-10-02649-f001:**
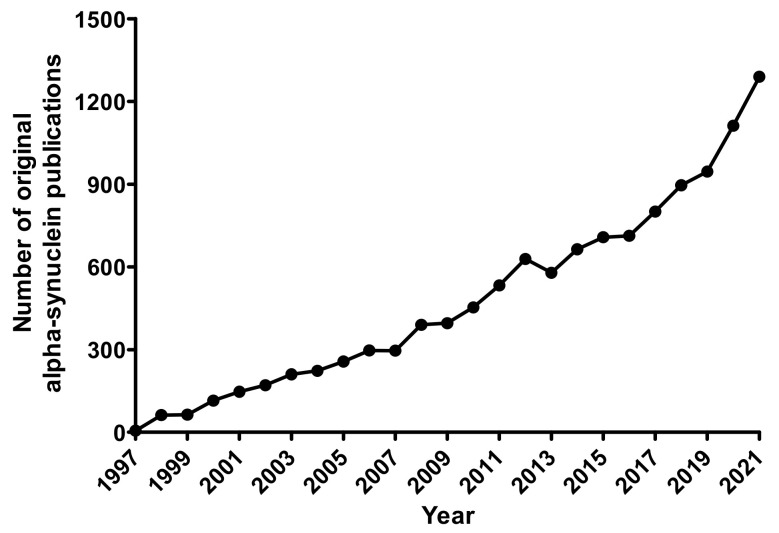
Number of original articles published per year on “alpha-synuclein” starting from 1997 when it was demonstrated that α-syn is involved in PD, up until 2021. Since 2020, over 1000 articles have been published in a year. Data was obtained from Scopus (https://www.scopus.com/, accessed on 10 January 2021). A search was made for the word “alpha-synuclein” within ‘Article title, Abstract, Keywords’. This was further refined to Articles (so as to exclude Reviews and Book Chapters), then the years were narrowed from 1997 to 2021. This generated a total of 11,959 results.

**Figure 2 biomedicines-10-02649-f002:**
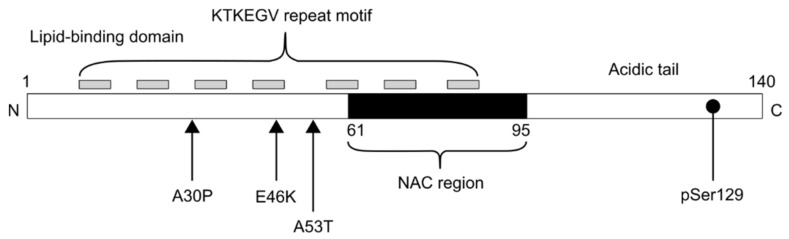
Structural diagram of alpha-synuclein protein. The N-terminal contains an amphiphilic region characterized by apolipoprotein repeat motif. The non-amyloid component (NAC) region is important in the oligomerization and aggregation of α-syn. The C-terminal domain is characterized by acidic residues Glu and Asp and assumes an inherently disordered conformation. Phosphorylation at Ser129 is a classical marker for aggregated α-syn.

**Table 1 biomedicines-10-02649-t001:** Examples of recent studies with the use of non-iPSCs models with different α-syn modifications. The definition of PFF and oligomeric forms of α-syn can vary from paper to paper. PFF—preformed fibrils.

Model	α-syn Modification	Outcomes of the Study	References
Cell lines			
SH-SY5Y	Overexpression	Recipient microglia suppressed autophagy caused by enhanced expression of miR-19a-3p in exosomes, via the phosphatase and tensin homolog/AKT/mTOR signaling pathway.	[26]
	Overexpression	miR-335 levels are reduced in PD models and patients. Its overexpression reduced inflammation induced by LPS stimulation or LRRK2 overexpression.	[27]
	A53T mutation	Reduced mitochondrial oxygen flow at maximum capacity.	[28]
	A53T mutation	Silencing of CK2α results in reduced phosphorylated α-syn at serine129 expression in cells with A53T mutation as well as functionality of dopaminergic neurons and ROS generation.	[29]
	PFFs	Micropinocytosis was suggested to be the main pathway of α-syn internalization into SH-SY5Y cells and differentiated neurons.	[30]
	PFFs	Higher secretion levels of nanoscopic α-syn aggregates were driven by disrupted protein homeostasis caused by PFF, but it does not lead to aggregation in the cells.	[31]
	Monomers, dimers, tetramers	α-syn dimer and tetramer internalization into the cell happened primarily through endocytosis. Aggregated α-syn from PFFs or oligomers displayed more prominent accumulation than monomers. Tetramer structures of α-syn showed more resistance to these processes, which suggested higher infectiousness of higher oligomeric states.	[32]
	Monomers, polymers,α-syn-119	1:4 ratio of α-syn -119:PQQ (pyrroloquinoline quinone) resulted in neuroprotective effect, showing antioxidant effect of PQQ. PQQ can change the secondary structure of α-syn, inhibiting oligomer formation induced by Cu(II).	[33]
PC12	Overexpression	Glutamine can enhance Hsp70 expression which is able to promote degradation of α-syn even in the presence of a proteasomal inhibitor.	[34]
	Overexpression	Increase in oligomerization and aggregation of α-syn might be the result of iron accumulation in neurons. The abnormal iron levels can be caused by higher levels of α-syn concentration in the cells.	[35]
	Oligomers and overexpression	Serotonin aldehyde oligomerizes α-syn in vivo and in vitro.	[36]
	A53T mutation	Ubiquitin proteasome system dysfunctions caused by α-syn in dopaminergic neurons	[37]
	PFFs	PC12 cell line is less resistant to α-syn cytotoxicity than primary hippocampal neurons.	[38]
	PFFs	α-syn fibril formation can be inhibited by hydroxytyrosol, and fibrils can be destabilized by hydrotyrosol.	[39]
	(MPP^+^)-treated cells	Low-intensity ultrasounds stimulation results in ROS generation inhibition in MPP+ treated cells, lowering levels of α-syn aggregation.	[40]
LUHMES	Overexpression	Transcriptome and proteasome analyses identified differential regulation of genes associated with PD. Vesicular transport and the lysosome were leading mechanisms.	[41]
	SNCA knockout	401 genes associated to the cell cycle had reduced expression after SNCA knockout in dopaminergic neurons.	[42]
	Overexpression	During drug screening, PDE1A inhibition showed the most effective results against α-syn toxicity.	[43]
	A30P mutation	Overexpression of WT and A30P m α-syn had a significant effect in DNA methylation of genes related to glutamate signaling and locomotor pathways	[44]
MN9D	Normal expression	α-syn accumulation was inhibited by suppression of prolonged adenosine A1 receptor activation.	[45]
	Normal expression	Damage induced by 6-OHDA can be reduced by icaritin (ICT). ICT increases SOD activity, TH expression, but decreased ROS production and α-syn expression.	[46]
BV2	A53T mutation	Polygala saponins fractions inhibited NLRP3 inflammasome by AMPK/mTOR and PINK1/parkin pathways, which contribute to the regulation of neuroinflammation decrease and neuronal death via mitophagy	[47]
	α-syn and MPP+ co-treatment	α-syn and MPP+ co-treatment induced activation of NLRP3 inflammasome.	[48]
	Monomers, oligomers	Monomeric α-syn promotes microglial inflammatory phenotype by ERK, NF-κB, and PPARγ pathways.	[49]
	α-syn -enriched conditioned media	Neuroinflammation caused by impairment in microglial autophagy is disrupted by α-syn on the Tlr4-dependent p38 and Akt-mTOR pathway.	[50]
	PFFs	α-syn fibrils caused a strong inflammatory response. Level of fibrilization is a main trait for its intake.	[51]
	A53T mutation	Norepinephrine release caused dopaminergic neuron viability disruption in the noradrenergic system.	[52]
Primary neurons	A53T α-syn T22N Rab7A mutations	Wild type Ras-related in brain 7 (Rab7) reduced α-syn decreased α-syn toxicity, e.g., oxidative stress, mitochondrial perturbations, and DNA damage.	[53]
	PFFs and α-syn E35K E46K E61K mutants	Lysophosphatidylcholine acyltransferase 1 regulates α-syn pathology. Utilization of α-syn E35K E46K E61K model.	[54]
	LRRK2 inhibition	α-syn localization at the presynaptic terminal is connected to the kinase activity of LRRK2.	[55]
	PFFs	α-syn aggregation induced by PFFs is not influenced by insulin-related signaling in primary dopamine neurons.	[56]
	PFFs	Tannic acid showed the best results in two-step screening for α-syn aggregation inhibitors.	[57]
	PFFs and AS69 protein	There is no change in PFF uptake in the presence to AS69 protein, that binds to α-syn, but AS69 decreases α-syn pathology.	[58]
	Monomer	GLP-1R-associated neuroprotective and neurotrophic cell signaling can be activated by GLP-1 (9–36).	[59]
Human neural stem cell line (ReNcell)	PFF or overexpression	α-syn and its aggregate degradation can happen with miR-7 use. miR-7 can also decrease α-syn expression.	[60]

**Table 2 biomedicines-10-02649-t002:** Examples of studies with the use of iPSCs models with different α-syn modifications. The definition of PFF and oligomeric forms of α-syn can vary from paper to paper. PFF—preformed fibrils.

iPSCs Model	α-syn Modification	Outcomes of the Study	References
iPSC- derived neurons	Overexpression	α-syn binds directly to the bTubIII and it is linked to the neuritic integrity in PD.	[111]
	A53T mutation	Higher levels (mRNA) of α-syn, early changes in expression of genes related to metabolism, differentiation/development, ion transport, cytoskeleton, extracellular matrix organization, and synaptogenesis.	[112]
	A53T mutation	Abnormal accumulation of α-syn disrupts mRNA stability in PD iPSC neurons, disturbs the decapping module in PD brain.	[113]
	A53T mutation and isogenic line	Increase in SNCA/α-syn can be enhanced by recombinant pro-cathepsin D.	[114]
	A53T mutation	Neuron degradation, protein aggregates, increase in protein synthesis.	[28]
	SNCA A53T and GBA1 mutation	ER fragmentation can be caused by the α-syn accumulation in midbrain neurons.	[115]
	LRRK2 mutation	Carbosilane dendrimers use can counteract abnormal α-syn accumulation.	[116]
	A53T mutation and PFFs	Generation of reliable humanized seeding model for pharmacological research.	[117]
	PFFs and ribbons	Spreading of α-syn fibrils and ribbons, aggregation of endogenous α-syn.	[118]
	PFFs	α-syn aggregation in the form of phosphorylated α-syn.	
iPSC-derived astrocytes	LRRK2 mutation	LRRK2 and GBA mutations in astrocytes contribute to PD development, manifesting several disease hallmarks.	[119]
	ATP13A2 mutation	ATP13A2 mutation in astrocytes results in α-syn accumulation in dopaminergic neurons and ATP13A2 deficiency compromises protective astrocytes function from α-syn aggregation.	[120]
	LRRK2 mutation	Astrocytes play role in dopaminergic cell death in PD pathogenesis, by dysfunctions in pathway of protein degradation.	[121]
	PFFs	Astrocytes and microglia revealed synchronous activity in processing α-syn aggregates.	[122]
	PFFs	Exposure of α-syn to PFFs leads to antigen presenting phenotype in astrocytes with upregulation of major histocompatibility complex and antigen molecules, while TNF-α activates pro-inflammatory pathway.	[123]
	PFFs	Astrocytic α-syn uptake can be limited by binding to clusterin. A-syn clearance can be improved with lower clusterin levels.	[124]
iPSCs-derived endothelial cells	PFFs	Generation of a substantia nigra brain chip, reproducing α-syn pathology in vivo during PFFs exposure.	[125]
iPSCs-derived oligodendrocytes and midbrain spheroids	Overexpression and A53T mutation	PD and MSA can affect oligodendrocytes in early cellular pathways and alterations. Epigenetic, genetic changes, and immune reactivity in MSA can be connected to each other by immune component triggered by α-syn.	[126]
Organoids (iPSCs)	Overexpression	SNCA triplication in midbrain organoids revealed pathological hallmarks of synucleinopathies in glial and neuronal cells.	[127]
	PINK1 mutation	2-Hydroxypropyl-β-Cyclodextrin treatment resulted in mitophagy improvement and better dopaminergic differentiation by protein levels modifications.	[128]
	PARKIN mutation	Mutation in PARKIN results in reduced IF activity.	[129]
	PINK1 mutation	Decreased amount of dopamine in vesicles, higher expression of α-syn.	[130]
	LRRK2 mutation	Midbrain organoids with LRRK2 mutation showed 3D pathological hallmarks of sporadic PD in patients. Thiol oxidoreductase functions are important in the LRRK2-associated PD development.	[131]
	APOE knockout	Lower levels of apoE induce aggregation of insoluble α-syn and phosphorylated α-syn, increased synapse loss, excess lipid droplet formation (hence GBA reduction and endo-lysosomal dysregulation).	[132]
	PFFs	Enteroendocrine cells are a key component of gut-brain hypothesis for the outcome and α-syn pathology development, and they show uptake and propagation of PFFs to neurons.	[133]
Organoids (ESC-derived)	GBA1 knockout and overexpression	β-sheet–rich α-syn aggregates can be the result of loss of glucocerebrosidase, linked with α-syn overexpression.	[134]
	DNAJC6 mutation	Loss of function in DNAJ6 gene results in α-syn aggregation caused by impairment in autophagy.	[135]

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
