# Peer review of "Cellular Models of Alpha-Synuclein Aggregation: What Have We Learned and Implications for Future Study"

_biomedicines, 2022, doi:10.3390/biomedicines10102649_

Round 1
Reviewer 1 Report
This is a comprehensive review of current knowledge of pathologies associated with alpha-synuclein aggregation. Emphasis is made on cellular models to achieve current progress.
Minor comments:
- a description of the structure of alpha-synuclein given in lines 49-71 would benefit from having a schematic representation of it;
- speaking about the disordered structure of alpha-synuclein, authors should mention the association of ID proteins with membrane-less organelles and the potential involvement of the latter in amyloidoses;
- the authors should provide the graphical abstract they refer to in line 71
- there are numerous duplicate references in the bibliography (for example, 41 and 83, 78 and 87…);
- expression “expressing the XXX mutation” is incorrect;
- the sentence in lines 192-194 is stylistically incorrect;
- the intended form of alpha-synuclein is not clearly indicated in all cases (see lines 521, for example).
Author Response
We thank the reviewer for agreeing to critically comment on our review and for their attention to detail in improving the manuscript. The response letter to review comments has been attached.

Reviewer 2 Report
In this manuscript, Albert and colleagues summarize cellular models of alpha-synuclein aggregation. Importantly, the authors offer insights into critical issues and future research directions in the field. The manuscript is overall balanced and well-written.
If possible, please summarize more recent developments and latest advances in 3D brain organoids as models of synucleinopathies.
I, or probably many readers, would like to read more about the future research directions of cell models for drug screening.
Author Response

(The authors gave the same response as above.)
